# Remote Sensing of Instantaneous Drought Stress at Canopy Level Using Sun-Induced Chlorophyll Fluorescence and Canopy Reflectance

**Simon De Cannière** [1,2,*], **Harry Vereecken** [2], **Pierre Defourny** [1] **and François Jonard** [1,2,3]

1   Earth and Life Institute, Université Catholique de Louvain, 1348 Ottignies-Louvain-la-Neuve, Belgium; pierre.defourny@uclouvain.be (P.D.); francois.jonard@uliege.be (F.J.)

2   Agrosphere (IBG-3), Institute of Bio- and Geosciences, Forschungszentrum Jülich GmbH, 52428 Jülich, Germany; h.vereecken@fz-juelich.de

3   Earth Observation and Ecosystem Modelling Laboratory, SPHERES Research Unit, Université de Liège, 4000 Liège, Belgium

\*   Correspondence: simon.decanniere@uclouvain.be

**Abstract:** Climate change amplifies the intensity and occurrence of dry periods leading to drought stress in vegetation. For monitoring vegetation stresses, sun-induced chlorophyll fluorescence (SIF) observations are a potential game-changer, as the SIF emission is mechanistically coupled to photosynthetic activity. Yet, the benefit of SIF for drought stress monitoring is not yet understood. This paper analyses the impact of drought stress on canopy-scale SIF emission and surface reflectance over a lettuce and mustard stand with continuous field spectrometer measurements. Here, the SIF measurements are linked to the plant's photosynthetic efficiency, whereas the surface reflectance can be used to monitor the canopy structure. The mustard canopy showed a reduction in the biochemical component of its SIF emission (the fluorescence emission efficiency at 760 nm—$\epsilon_{760}$) as a reaction to drought stress, whereas its structural component (the Fluorescence Correction Vegetation Index— FCVI) barely showed a reaction. The lettuce canopy showed both an increase in the variability of its surface reflectance at a sub-daily scale and a decrease in $\epsilon_{760}$ during a drought stress event. These reactions occurred simultaneously, suggesting that sun-induced chlorophyll fluorescence and reflectance-based indices sensitive to the canopy structure provide complementary information. The intensity of these reactions depend on both the soil water availability and the atmospheric water demand. This paper highlights the potential for SIF from the upcoming FLuorescence EXplorer (FLEX) satellite to provide a unique insight on the plant's water status. At the same time, data on the canopy reflectance with a sub-daily temporal resolution are a promising additional stress indicator for certain species.

**Keywords:** SIF; photosynthesis; photochemical reflectance index; isohydricity; non-photochemical quenching; water limitation; light limitation; soil water availability; vapour pressure deficit; vegetation index

## 1. Introduction

Over the last 50 years, remote sensing has proven its value for large-scale drought stress monitoring. Classically, remote sensing monitors the greenness of the vegetation. These methods provide an insight on stresses that a plant has suffered during its growing season [1], but it gives little information on the plant health at the moment the measurement was taken. Other techniques focus on the soil moisture content and on the meteorological variables [2], but they do not consider the wide variety in stress responses across ecosystems. A first attempt in monitoring the instantaneous plant stress came through evapotranspiration estimates, based on thermal remote sensing. In order to link thermal

measurements to evapotranspiration, it is necessary to have accurate data on various variables including the leaf and the air temperature, which is a major obstacle for interpreting thermal infrared data [3]. A more recent technique for measuring instantaneous stress uses passive microwave emission to calculate the vegetation optical depth, which is linked to the vegetation water content [4] or the turgor pressure [5]. A game changer in stress monitoring has been the introduction of the sun-induced chlorophyll fluorescence (SIF), a remote sensing signal sensitive to the photosynthetic activity of a plant [6] (Mohammed et al., 2019). Photosynthesis is particularly sensitive to drought conditions; since these induce stomatal closure to reduce the $CO_2$ uptake, the energy demand by the dark reactions is lowered. As a result, there is a need for an alternative energy sink, which is provided for by a series of reactions, collectively known as non-photochemical quenching (NPQ), which increases when the $CO_2$ uptake lowers. This, in response, also alters the SIF emission [7]. Through the stomata that are still open, plants lose water and therefore show a decrease in their turgor pressure [8]. This leads to changes in the leaf angle [9], changing the canopy reflectance, especially in the near-infrared [10]. This makes high-temporal resolution observations of near-infrared reflectance another promising indicator for instantaneous stress monitoring.

Since its discovery [11], chlorophyll fluorescence has proven its value for monitoring of photosynthetic activity, mostly through the form of pulse-amplitude modulation (PAM; [12]). Here, a leaf is illuminated with artificial light pulses, stimulating its chlorophyll molecules, after which they send out fluorescent light as a reaction. This fluorescent reaction informs the stress status. Specifically, stress lowers the actual PSII efficiency (YII), as the demand for energy in the photosynthetic electron transport chain lowers. As an alternative energy sink, NPQ increases. The increase in NPQ and decrease in Y(II) have a downstream effect on the emission of the steady-state fluorescence emission (Fs/F0). The latter variable describes the emission of fluorescence by a chlorophyll molecule under light-incubated conditions. Fs/F0 is conceptually similar to SIF, which is the steady-state emission of chlorophyll fluorescence measured by a spectrometer and uses the sun as illumination source. A drought-induced decrease in Fs/F0 has been observed over various species [13].

SIF provides a fluorescence signal from the leaf to the ecosystem scale. Since SIF only comprises of 1–2% of the absorbed energy, its retrieval is limited to solar and atmospheric absorption lines [14]. Canopy-scale SIF is typically retrieved at the $O_2$-A and $O_2$-B atmospheric absorption bands, situated at 760 nm and 687 nm, respectively. Their corresponding $SIF_\lambda$ observations are referred to as SIFA and SIFB. Canopy-scale $SIF_\lambda$ is expected to behave differently at these wavelengths. First, the relative contribution of photosystem I (PSI) to the $SIF_\lambda$ emission is more important for SIFA compared to SIFB. Second, 687 nm is within the absorption spectrum of chlorophyll, causing SIFB to be prone to reabsorption by another chlorophyll molecule. This is not the case of SIFA. Third, the re-emitted photon is scattered around in the canopy, which is wavelength-specific behaviour [15]. Another remote sensing indicator that is linked to the plant biochemistry is the photochemical reflectance (PRI; [16]), which is inversely correlated to the NPQ. Therefore, SIF and PRI are expected to provide similar, yet complementary information on the plant's photosynthetic activity.

Within the plant kingdom, various drought survival strategies exist. These strategies break down into two broad categories, isohydric and anisohydric, which exist alongside each other on a spectrum [17]. Isohydric plants tend to have a risk-avoidant strategy, closing their stomata immediately in cases of a water shortage. This leads to a reduced growth in case of a drought stress. The anisohydric strategy is a higher risk strategy, in which a plant keeps its stomata open, maintaining its growth rate while suffering high water losses through transpiration [18], thereby showing larger variations in turgor pressure [17]. Plants following a more anisohydric strategy are expected to show a fierce reaction in their canopy structure compared to more isohydric plants.

Because of its mechanistic link to photosynthesis, canopy-scale SIF is typically linearly linked to ecosystem-scale photosynthesis, quantified by the Gross Primary Productivity (GPP; [19,20]), but the link between SIF and GPP is less straightforward in cases of drought

stress. A field-scale SIF-GPP relationship could be finetuned during drought periods using PRI data [21]. Field-scale studies have pointed out drought stress at both the structural and biochemical level [22]. Both stress-induced changes in the leaf biochemistry and in the canopy structure affect both SIFA and SIFB [23]. Due to the sensitivity of the biochemical component to drought stress, SIF proved to be an interesting input for a crop growth model [24].

This paper describes the reaction of SIFA and SIFB to increasing drought stress conditions by monitoring the SIF emission and canopy reflectance of a yellow mustard (Sinapsis alba) and a common lettuce (Lactuca sativa) canopy continuously over their growing season, and thus capturing a variety of stress conditions. In addition, the near-infrared reflectance is monitored to estimate short-term variations in canopy structure. Yellow mustard and common lettuce differ greatly in their degree of isohydricity, in which the mustard is more isohydric than the lettuce. We expect the lettuce to show a more expressed structural reaction because it tends to lose more turgor pressure given its anisohydric nature. The mustard's more isohydric nature should force its stomata to close more quickly, and thereby causing a reaction at the level of the leaf biochemistry.

## 2. Materials and Methods

### 2.1. Site Description and Experimental Design

The experiment took place at the remote sensing field test facility of Forschungszentrum Jülich in Selhausen (Niederzier, Germany). Two 2 by 2 metre wooden boxes were built with a soil depth of 40 cm (Figures 1 and 2).

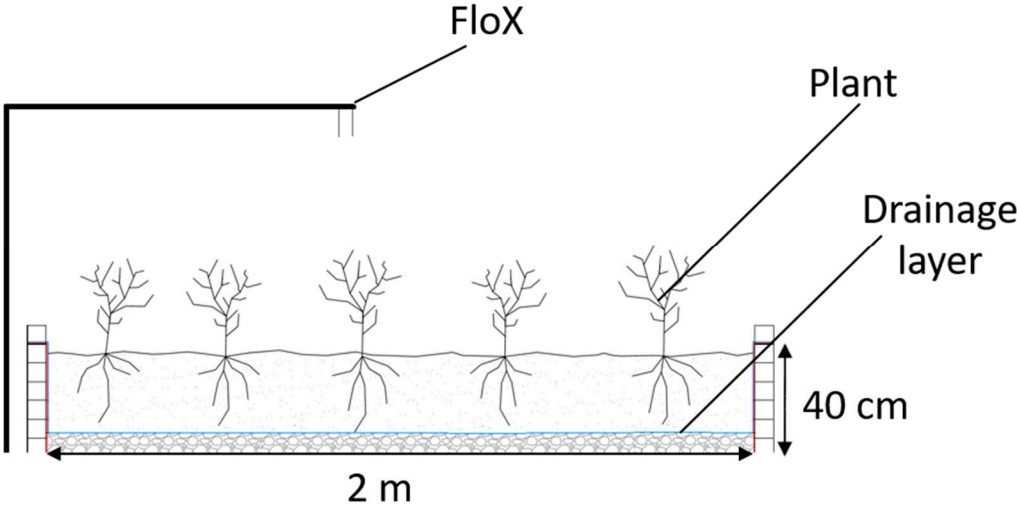

**Figure 1.** Schematic view of the box design.

The boxes were filled with the soil from the site, containing 13% sand, 70% silt and 17% clay, according to the USDA textural classification [25]. The soil was placed on top of a gravel layer, with geotextile separating the silty loam from the gravel layer. This setup allows standing water to drain away, limiting its water storage capacity. Gravel and stones were removed from the soil.

A tripod with a hyperspectral instrument (Fluorescence box—FloX) was placed in between the boxes. The tripod was installed in such a way that the spectrometer could pivot between the boxes. During the vegetation development phase, identical irrigation schemes were applied to both boxes. After the leaf development phase, the boxes were subject to different irrigation regimes, causing one box to become significantly dryer compared to the other. By default, the FloX instrument was installed to measure the dry box. During specific days, the cross-arm on the tripod was turned for the FloX to measure over the reference box.

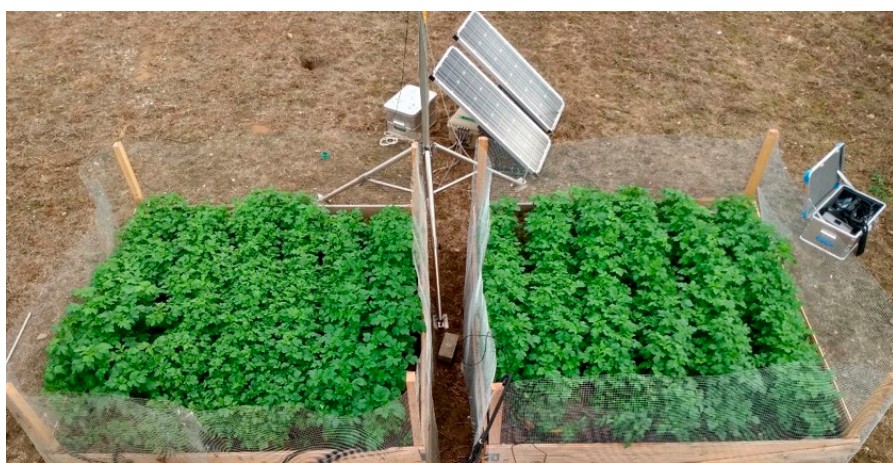

**Figure 2.** Picture of the boxes during the mustard experiment.

The test facility was located next to the Integrated Carbon Observatory System (ICOS) and TERENO (Terrestrial Environmental Observatories) test site at Selhausen, from which data on air humidity and air temperature were used. Soil moisture, soil temperature and soil water potential (matric potential) were monitored in the plant boxes themselves. Table 1 gives an overview of the sensors used. The soil water availability was estimated by means of the soil water potential, expressed in pressure heads (h). These were converted to a pF value (Equation (1)). Soil moisture and soil water potential provide complementary information. Soil moisture is more commonly used in modelling of the hydrological cycle as it is easier to measure at large scale [26], whereas soil water potential corresponds more closely to the water the plant can extract from the soil [27] (Robbins and Dinneny, 2018).

$$pF = -\log(h) \tag{1}$$

**Table 1.** Overview of equipment used during the measurement campaigns.

| Variable | Symbol | Unit | Instrument |
|---|---|---|---|
| Air Temperature | $T_{Air}$ | $°C$ | HPM45C, Vaisala Inc., Helsinki, Finland |
| Air Humidity | RH | % | HPM45C, Vaisala Inc., Helsinki, Finland |
| Vapour pressure deficit | VPD | kPa | HPM45C, Vaisala Inc., Helsinki, Finland |
| Precipitation | Prec | mm | Thies Clima tipping bucket, Ecotech, Bonn, Germany |
| Photosynthetically active radiation | PAR | $Wm^{-2}$ | FloX, JB Hyperspectral, Düsseldorf, Germany |
| Soil moisture | SM | $cm^3cm^{-3}$ | 5TE, Meter Environment, München, Germany |
| Soil water potential | SWP | kPa | Teros-21, Meter Environment, München, Germany |

Two measurement campaigns were conducted. The mustard was sown on May 29th (DOY 149), 2021 and harvested on August 2nd (DOY 214), 2021. The lettuce was planted (not sown) on August 7th (DOY 219), 2021 and harvested on September 22nd (DOY 265), 2021.

### 2.2. Leaf-Scale Measurements of Photosynthesis

In order to link the SIF signal to photosynthesis, leaf-scale active chlorophyll fluorescence measurements were taken with the portable PAM 2500 chlorophyll fluorimeter (Heinz Walz GmbH, Effentrich, Germany), to gain information on the efficiency of the PSII (Y(II)) and on the steady-state fluorescence (Fs/F0). Pulse-amplitude modulation (PAM) measurements were taken on sunny days, preferably, and multiple measurements were taken throughout the day.

### 2.3. Processing of Field Spectrometer Data

The SIF data were taken using a Fluorescence Box (FloX; JB hyperspectral, Düsseldorf, Germany), which combines two field spectrometers, retrieving in both the $O_2$-A and $O_2$-B

band, respectively, and situated at 760 and 687 nm using the improve Fraunhofer Depth Line method (iFLD; [28]). The spectrometers have a spectral sampling rate of 0.17 nm and 0.65 nm, respectively. Their respective spectral resolutions are 0.3 nm and 1.5 nm. The iFLD method compares the up- and downwelling radiances inside the oxygen absorption bands to the radiance just outside the absorption bands, resulting in an estimate for the SIFA and SIFB fluorescence [29]. In addition to the fluorescence, the FloX measures the hyperspectral surface reflectance in the region of 400–900 nm. The FloX spectrometer took data from 7 AM until 7 PM UTC+1 with a temporal resolution of one minute. Outliers in the fluorescence data, diverging more than 3 standard deviations from the daily mean, were removed. Then, both the fluorescence, irradiance and reflectance data were averaged to a temporal resolution of 30 min, reducing the noise in the time series.

### 2.4. Monitoring Leaf Biochemistry with Field Spectrometer Data

The canopy-scale SIF is not merely the sum of the SIF emissions of the individual leaves. Instead, considering the canopy scattering and reabsorption is vital when moving from leaf-level to canopy-level SIF, the behaviour of scattering and reabsorption are wavelength-specific. Consequently, SIF at a certain wavelength $\lambda$ observed in a certain direction $\omega$ ($SIF_{\lambda,\omega}$) is affected by two wavelength-specific probabilities; the fluorescence emission efficiency $\epsilon_\lambda$ and the photon's escape probability $f_{esc,\lambda,\omega}$ (Equation (2)). Both $f_{esc,\lambda,\omega}$ and $\epsilon_\lambda$ are expected to be reactive to water availability. This paper aims at describing these two factors and their relation to drought stress.

$$SIF_{\lambda,\omega} = PAR \cdot fPAR_{Chl} \cdot \epsilon_\lambda \cdot f_{esc,\,\lambda,\omega} \qquad (2)$$

Isolating $\epsilon_\lambda$ from the Equation (1) allows extracting the biochemical information from the spectrometer-based $SIF_{\lambda,\omega}$. Yang et al., (2020) [30] developed the Fluorescence Correction Vegetation Index (FCVI; Equation (3)), which is a reflectance-based surrogate factor for the canopy absorption and canopy scattering. This index is formulated as the difference between the near-infrared reflectance $R_{NIR}$ between 767 and 773 nm and the broadband visible reflectance $R_{VIS}$ between 400 and 700 nm, assuming that the reflectance observations are taken from the same direction as the SIF measurements. This approach is only valid in the absence of any reabsorption, which is why it is only meaningful for SIFA.

$$FCVI = R_{NIR} - R_{VIS} \approx f_{esc,760} \cdot fPAR_{Chl} \qquad (3)$$

The biochemistry can be assessed with three variables: fluorescence emission efficiency at 760 nm ($\epsilon_{760}$: Equation (4)), the SIFB yield (SIFBY: Equation (5)) and the photochemical reflectance index (PRI: Equation (6)). The $\epsilon_{760}$ contains the SIFA normalized by the PAR and FCVI and integrated over a hemispherical space. The PRI is a reflectance-based, narrow-band vegetation index, related to the de-epoxydation of xanthophyll, which is linked to the NPQ. A more negative PRI should indicate a higher NPQ [16], and therefore a higher stress level. Similar to SIF, PRI is affected by a biochemical and a structural component [31]. The epoxidation of xanthophyll causes only a change in very limited parts of the absorption spectrum, which is why a bandwidth of 2 nm is chosen in Equation (6).

$$\epsilon_{760} = \frac{\pi \cdot SIFA}{FCVI \cdot PAR} \qquad (4)$$

$$SIFBY = \frac{SIFB}{PAR} \qquad (5)$$

$$PRI = \frac{R_{531} - R_{570}}{R_{531} + R_{570}} \qquad (6)$$

To assess the sensitivity to PAR of the leaf biochemistry, the daily correlation coefficient $\rho$ was calculated. To do so, the Pearson's correlation coefficient between the $\epsilon_{760}$ and SIBY was used on one hand and, on the other hand, PAR was calculated for each day individually,

resulting in a daily correlation coefficient. The daily correlation coefficient only considered data taken between 9 AM and 3 PM, since light-limited photosynthesis is expected in early morning or late afternoon, due to the lower solar irradiations at these times of the day.

### 2.5. Monitoring Canopy Structure with Field Spectrometer Data

The canopy structure was monitored by means of the FCVI. Canopy structure contains two elements: the leaf area index (LAI), which depends on the plant's phenology, and the leaf angle. The leaf angle is subject to sub-daily changes. A large variation in leaf angle is expected to translate into a large diurnal variation in FCVI. Trends in the FCVI are linked to change in LAI due to vegetation growth.

## 3. Results

### 3.1. Description of the Meteorological Conditions during the Growing Seasons

The mustard grew mainly in June and July, 2021. The growing season had a precipitation of 101.7 mm and mean temperature of 18.34 °C. The mustard reached its adult phase in July, a month with a mean temperature of 17.9 °C. On 16 July 2021 (DOY 197), the experiment was struck by unusual amounts of rainfall, so much so that the equipment could not keep up with the precipitation. However, the high rainfall explains the sharp peak in soil moisture on that day. The period after, between DOY 198 and DOY 205, showed a clear decrease in soil moisture.

The lettuce experiment took place in the late summer, with both lower precipitations (49.3 mm) and lower temperatures (16.9 °C), compared to the mustard experiment (Figure 3). During the 2021 lettuce experiment, more irrigation was applied to the reference box. These irrigation events resulted in slight increases in soil water content.

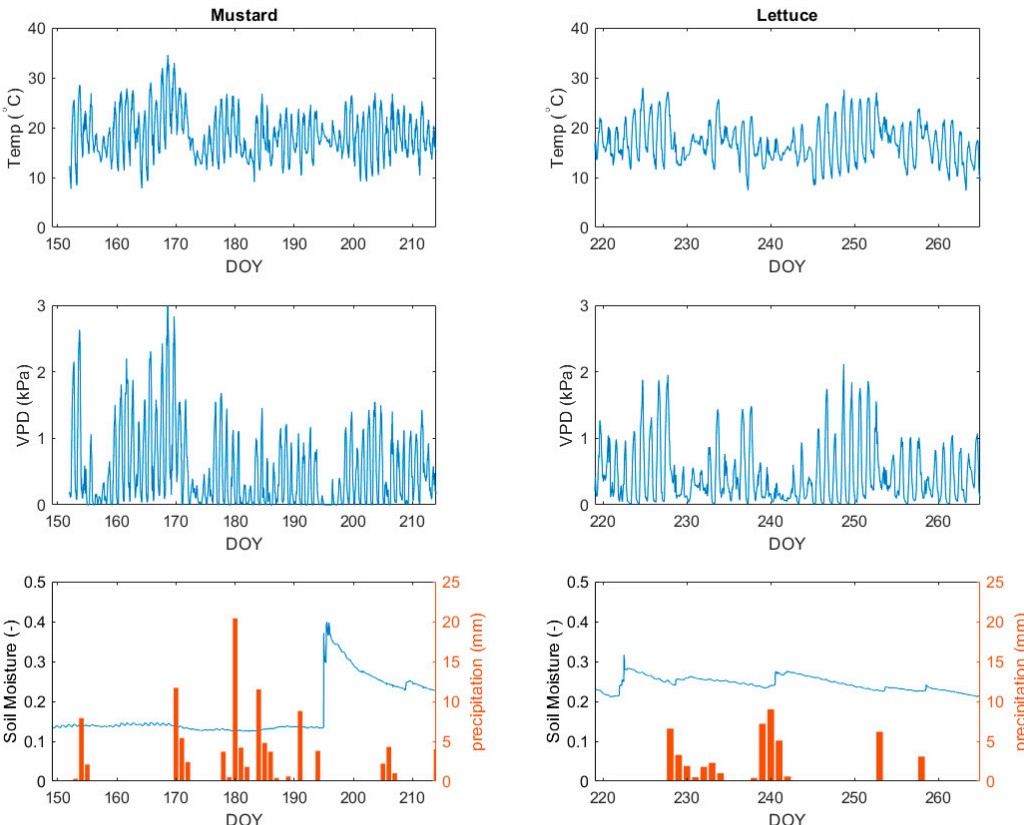

**Figure 3.** Description of the meteorological conditions during the growing season for mustard and lettuce.

### 3.2. Canopy Structure and SIF Emission at Sub-Daily Scale during Different Environmental Conditions

To illustrate the behaviour of the biochemical variables $\epsilon_{760}$ and PRI, as well as the behaviour of the structural variable FCVI at the diurnal level, two days of the lettuce

experiment and two days of the mustard experiment are plotted (Figure 4). These days had very different weather and soil conditions, corresponding to low and high water stress. In the lettuce dataset, the PRI remained constant over the unstressed day, whereas the $\epsilon_{760}$ showed a bell-shaped pattern. For the stressed day, the shape of the $\epsilon_{760}$ reverses to a valley shape. A similar valley-shaped pattern is visible for the PRI on that day. The biochemistry-related variables $\epsilon_{760}$ or PRI to stress between the lettuce and the mustard plant show a similar behaviour; both turn into a valley-like shape during the stressed period. The FCVI does however show a different reaction between the two species. Whereas the lettuce FCVI shows a bell shape during stressed days over the lettuce canopy, the FCVI barely shows any sensitivity to stress.

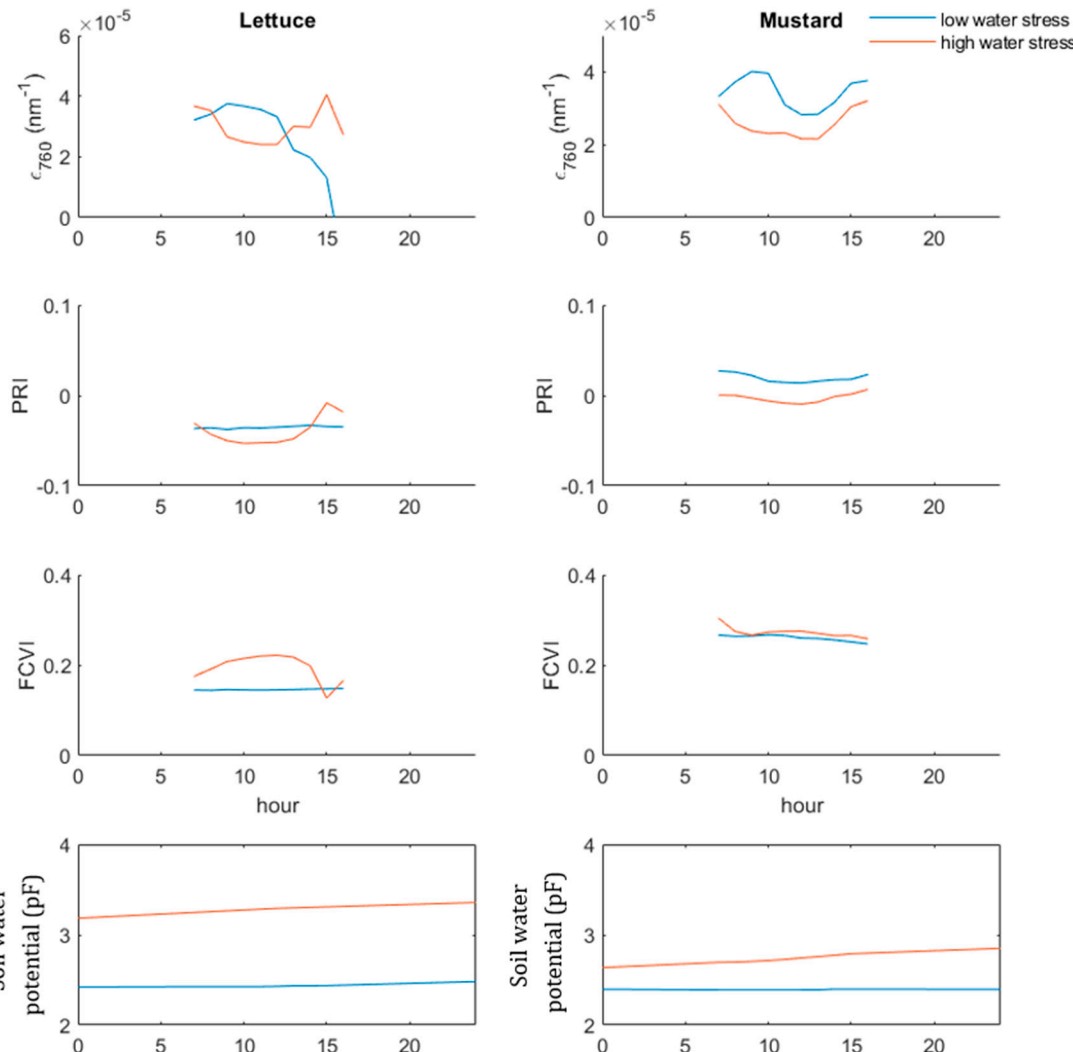

**Figure 4.** Behavior of $\epsilon_{760}$, PRI and FCVI for three days during the lettuce experiment. For the lettuce dataset, DOY 244 was the unstressed day (blue line), DOY 247 the stressed one (red line). For the mustard dataset, DOY 198 was the unstressed day (blue line), DOY 202 the stressed one (red line). The water stress conditions are indicated with the soil water potential.

### 3.3. Reaction of Structural and Biochemical Variables Daily Mean Value to Increasing Stress Level

In addition to the changes in the diurnal behavior, the daily mean value of the mustard biochemical variables changes in function of the stress intensity (Figure 5). This research considers a plant stressed if it suffers from both low soil water availability and from a high atmospheric demand. Between DOY 201 and DOY 209, the plants experienced a drought stress, through a combination of high PAR, VPD and pF values. The period between DOY 201 and 209 shows a decrease in $\epsilon_{760}$, compared to the period between DOY 196 and DOY

202. After DOY 209, the $\epsilon_{760}$ increased to its pre-stress levels. Like the $\epsilon_{760}$, the PRI showed a decrease between DOY 201 and DOY 204. At DOY 205, corresponding to a cloudy, colder day, the PRI jumped back up. This reaction was not observed with the $\epsilon_{760}$. It is interesting to note that a dry soil (i.e., high pF values), combined with low VPD and PAR values, does not induce a decrease in $\epsilon_{760}$, as seen in DOY 212 and DOY 213. During the mustard experiment, the FCVI hovered around 0.27 and showed little same-day variation. Unlike SIFA, SIFB shows very little reactivity to water limitation. Consequently, SIFA and SIFB are further apart during stressed days compared to the unstressed days.

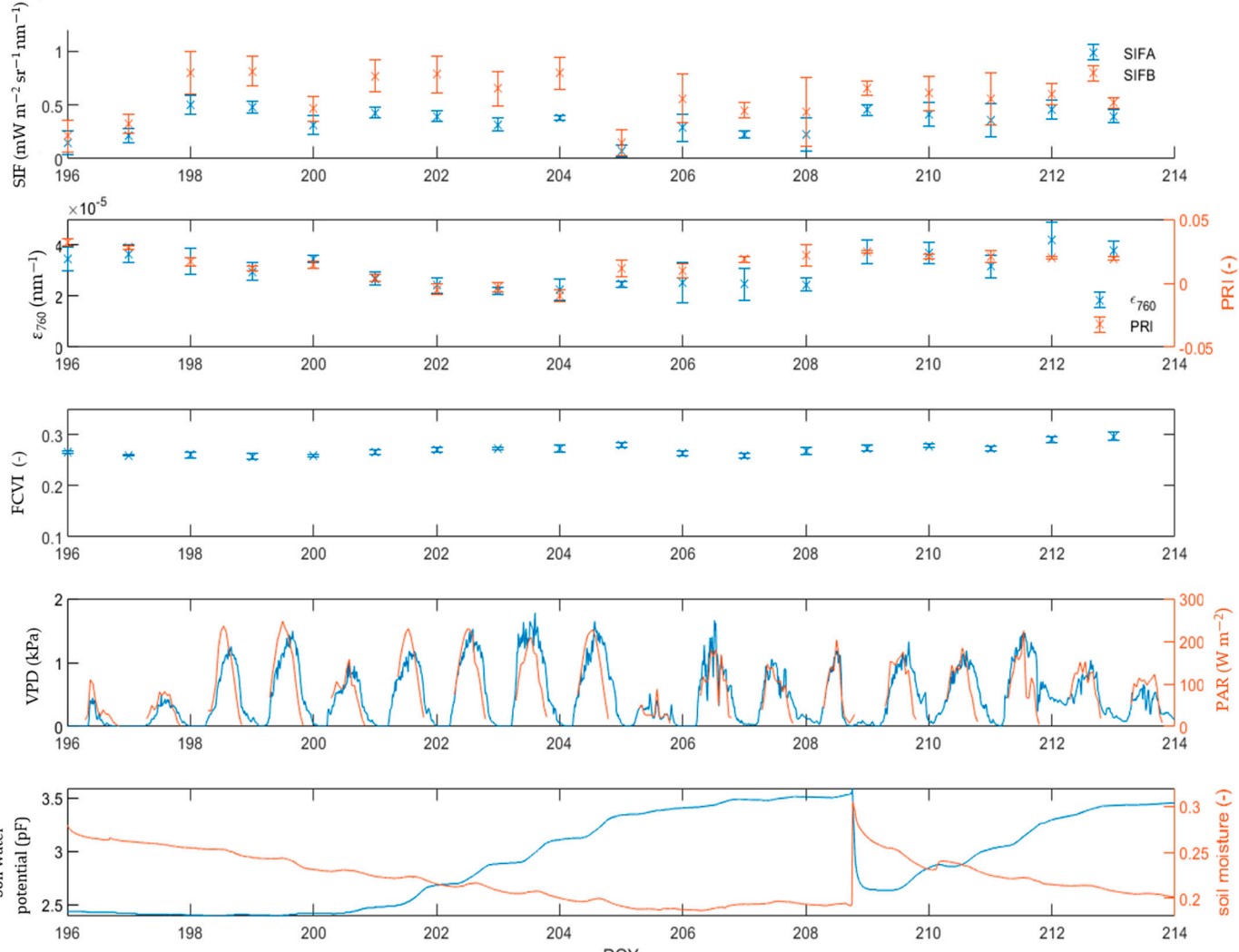

**Figure 5.** Evolution of SIFA, SIFB, FCVI, $\epsilon_{760}$ and PRI for the 2021 mustard campaign. The drought conditions are given by the VPD, PAR, soil water potential and soil moisture.

In the lettuce dataset, both the PRI and FCVI showed an upward trend during the growing season, explained by vegetation growth. The $\epsilon_{760}$ did not show such trend. Similar to the observations of the mustard dataset, lettuce biochemical variables ($\epsilon_{760}$, PRI) also show a decrease during drought events (Figure 6), albeit not as expressed as with the mustard dataset. The decreased values of $\epsilon_{760}$ were found between DOY 246 and DOY 251. During that same period, the PRI showed a slight decrease, before retaking its upward trend. Interestingly, the standard deviation of FCVI increased in days with high VDP, indicating that the lettuce plant shows major drought-induced, short-term variability in its canopy structure. The increased daily standard deviation in FCVI is prevalent in the period between 245 and 251, and it decreases in the days after. The mustard FCVI (Figure 5) did not show such behaviour.

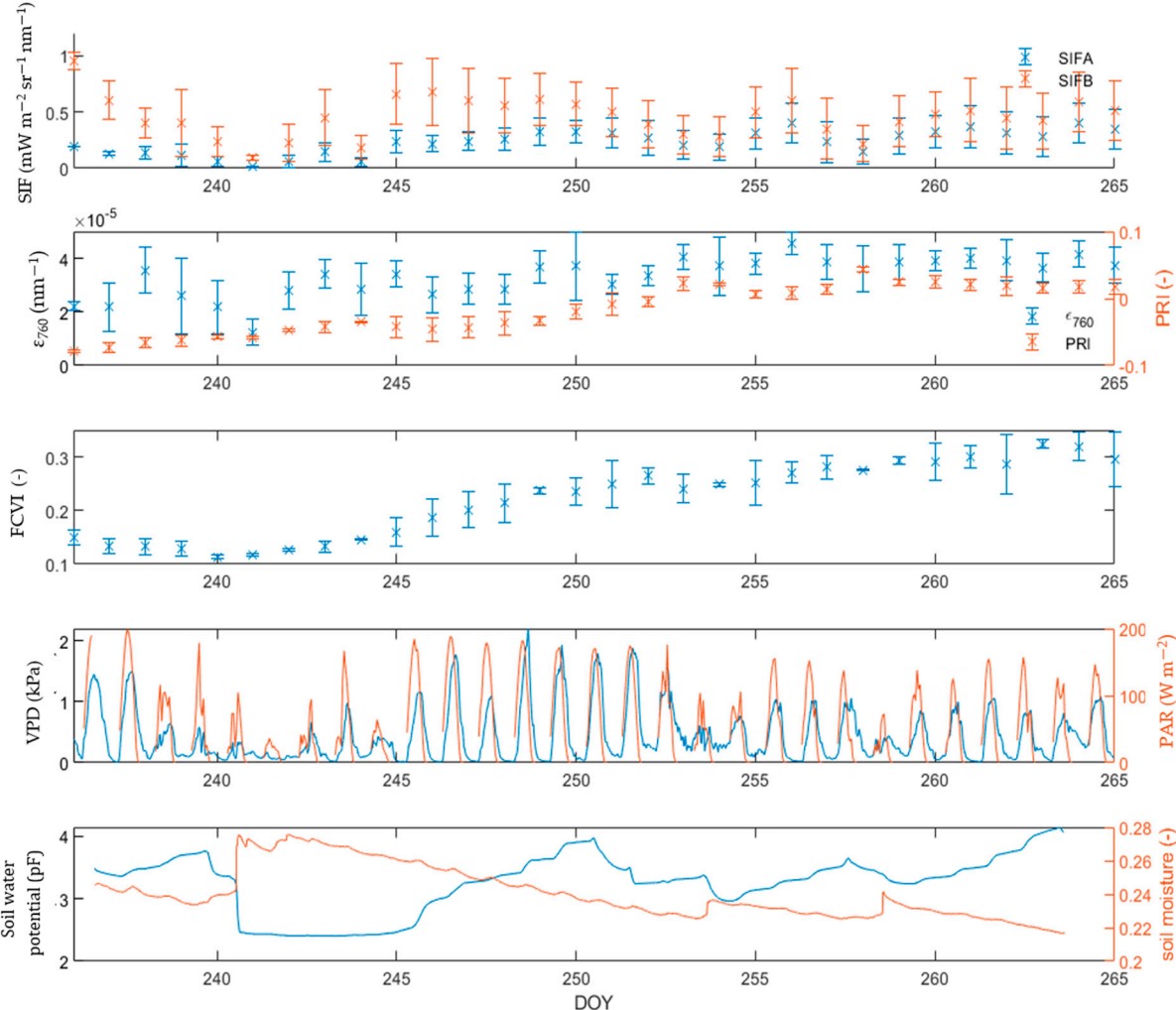

**Figure 6.** Evolution of SIFA, SIFB, $\epsilon_{760}$, FCVI, PRI for the lettuce campaign. The drought conditions are given by the VPD, PAR, soil water potential and soil moisture.

### 3.4. Relationship between SIF and PAR

Similar to light response curves used in photosynthesis research [32], the SIF-PAR curves were fitted with the Michaelis–Menten function (Equation (7)), in which the parameters $SIF_{max}$ and Km are fitting parameters. This equation suggests that an increase in PAR leads to an increase in SIF emission, up until a certain degree of saturation. The $\epsilon_{760}$-PAR curve was fitted using a second order polynomial function with the mustard reaching its maximum at PAR = 90 W/m$^2$ and lettuce reaching its maximum at PAR = 97 W/m$^2$. The lettuce $\epsilon_{760}$-PAR plot is more curved compared to its mustard counterpart (Figure 7). The SIFA-PAR curve for the lettuce shows a lot more scatter compared to its counterpart over the mustard canopy. The SIFB-PAR plots show a near-linear relationship. There is little difference in the SIFB-PAR plots between the two crops. The lower $R^2$ values for the SIFA-PAR compared to the SIFB-PAR plots suggest that there is more non-PAR related information in the SIFA compared to the SIFB.

$$SIF = \frac{SIF_{max} \cdot PAR}{Km + PAR} \tag{7}$$

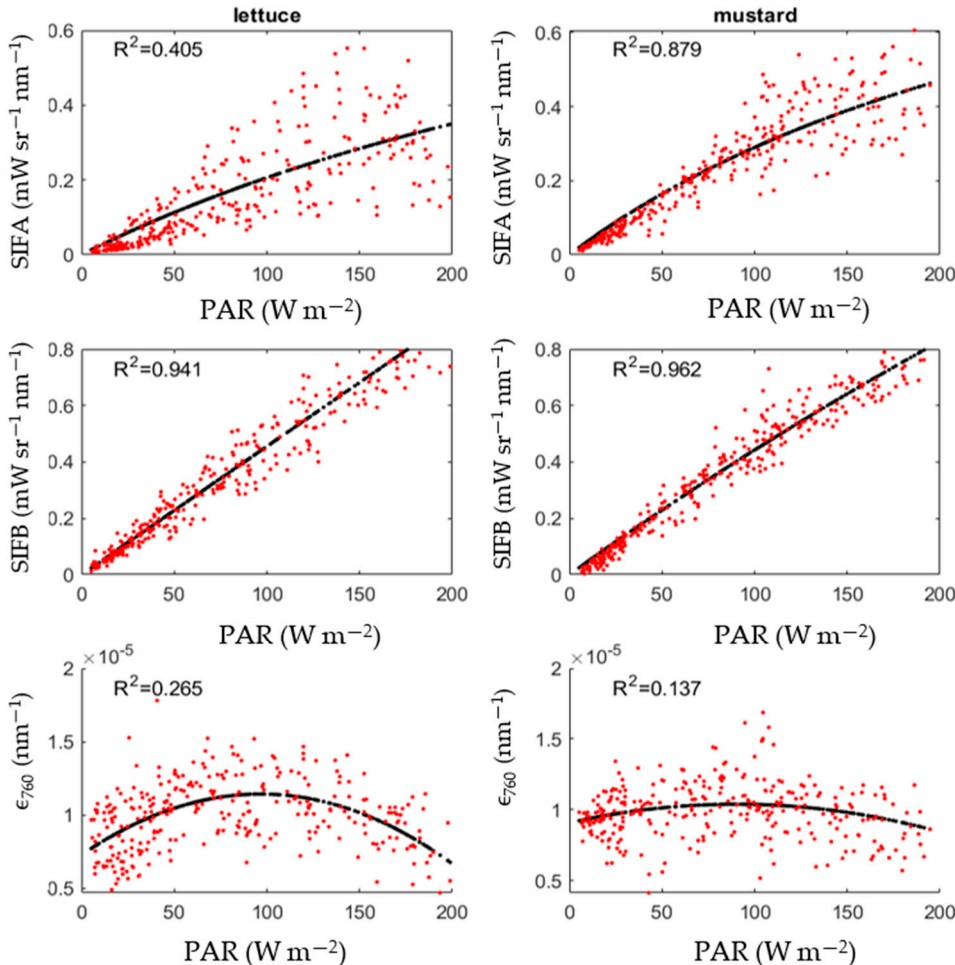

**Figure 7.** Fitted (black lines) and observed (red dots) values of SIFA, SIFB and $\epsilon_{760}$ in function of PAR.

### 3.5. Response of $\epsilon_{760}$ and SIFBY to Different Light Intensities under Light and Water Limitation

Figure 8 shows the daily correlation coefficient between PAR and two SIF related variables: $\epsilon_{760}$ and SIFBY for both the lettuce and mustard datasets. They each show a different behaviour. The daily $\rho$ between $\epsilon_{760}$ and PAR show a high, positive $\rho$ on days with clear light-limitation values (e.g., between DOY 196 or DOY 245). This correlation switches to an anti-correlation in the case of water limitation, as seen between DOY 198 and DOY 205 for the mustard dataset and between DOY 246 and DOY 251 for the lettuce dataset. This negative $\rho$ is quick to switch to a positive $\rho$ on days with a lower VPD, or on days with a lower pF value. Examples are the increased $\rho$ between DOY 206 and DOY 215 for the mustard dataset and between DOY 252 and DOY 260 for the lettuce dataset. The correlation between SIFBY and PAR remained between 0.5 and 1 for almost the entire growing season for both crops. A decrease in the SIFBY-PAR correlation is found between DOY 199 and DOY 205 for the mustard dataset, which corresponds to days with intense water limitation. The period between DOY 205 and DOY 209 still has a low soil water availability, but the period is cloudier, lowering the PAR and therefore putting the plant in a light-limited regime for large parts of the day. The anticorrelation between PAR and $\epsilon_{760}$ is the most expressed on DOY 203 and DOY 204, corresponding to the days with the highest pF values. Similarly, a drop in the $\rho$ between SIFBY and PAR is found on DOY 250 and DOY 251 for the lettuce dataset, corresponding to two days with clear water limitation.

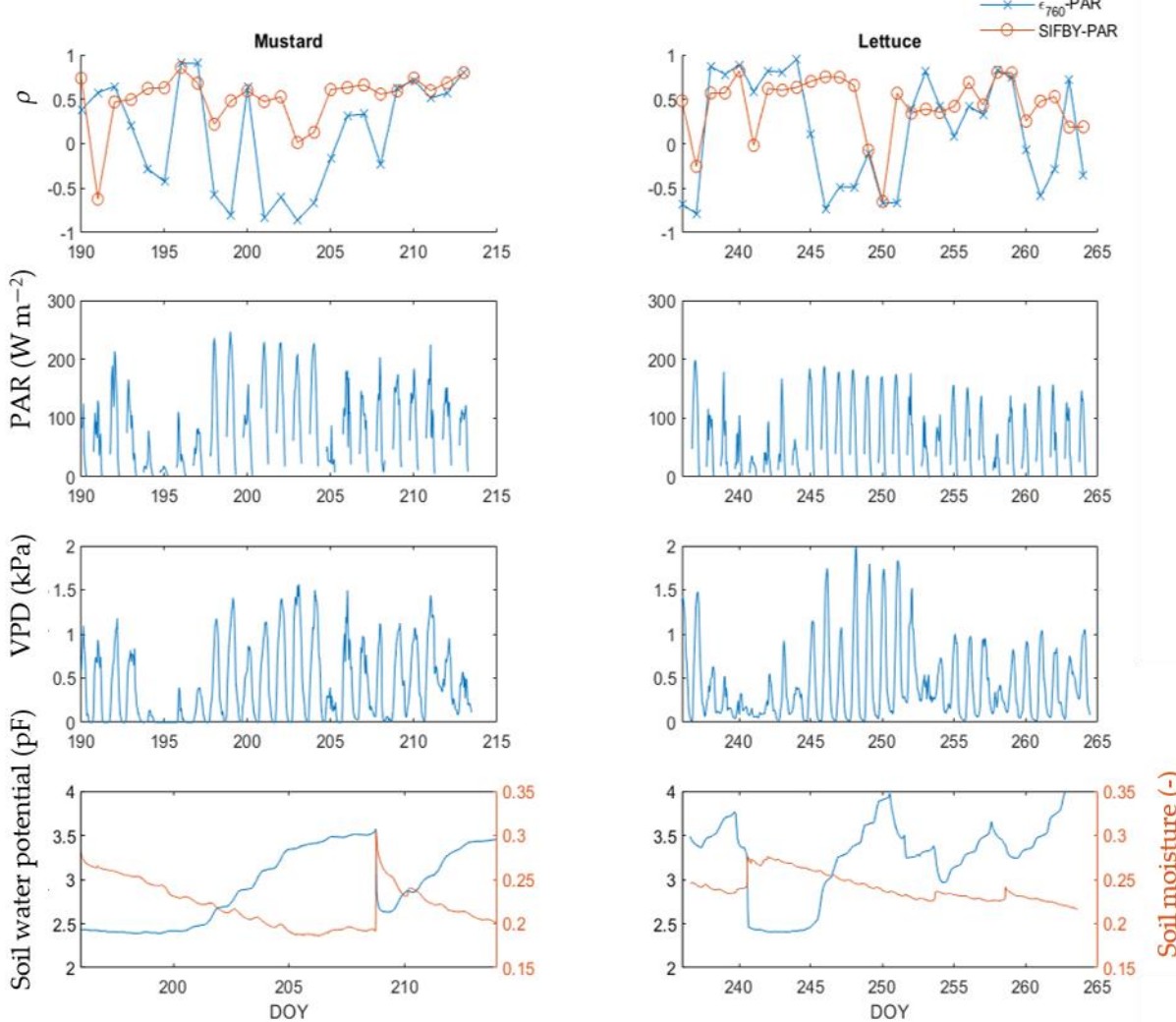

**Figure 8.** Daily correlation coefficient of $\epsilon_{760}$, SIFBY and FCVI to PAR for the mustard and lettuce experiments during their adult phases.

### 3.6. Linking PAM and FloX Measurements

Figure 9 shows a series of PAM measurements, taken on DOY 264 and DOY 271 during the lettuce experiment. DOY 264 was a sunnier day, with a more expressed water limitation. Two PAM parameters that were put special emphasis on were steady-state fluorescence Fs/F0, linked to the SIF, and the PSII effective quantum yield Y(II), linked to the PRI. The PAM data showed large variability, expressed by their standard deviation. This large variability is linked to changes in photosynthetic activity at the leaf level, due differences in illumination conditions, for example. During both DOY 264 and DOY 271, PRI and Y(II) followed a similar pattern, with their minimal values around between 11 h and 14 h. Linking Fs/F0 values to $\epsilon_{760}$ values is harder due to the large standard deviation of the Fs/F0 measurements. Both the Fs/F0 and $\epsilon_{760}$ values did not change significantly during either DOY 264 or DOY 271. The FloX measurements of $\epsilon_{760}$ between the stress box and the reference box did not differ significantly for DOY 264 and DOY 271. The PRI was significantly higher in the reference box on DOY 264, indicating higher stress conditions in the stressed box compared to the reference box.

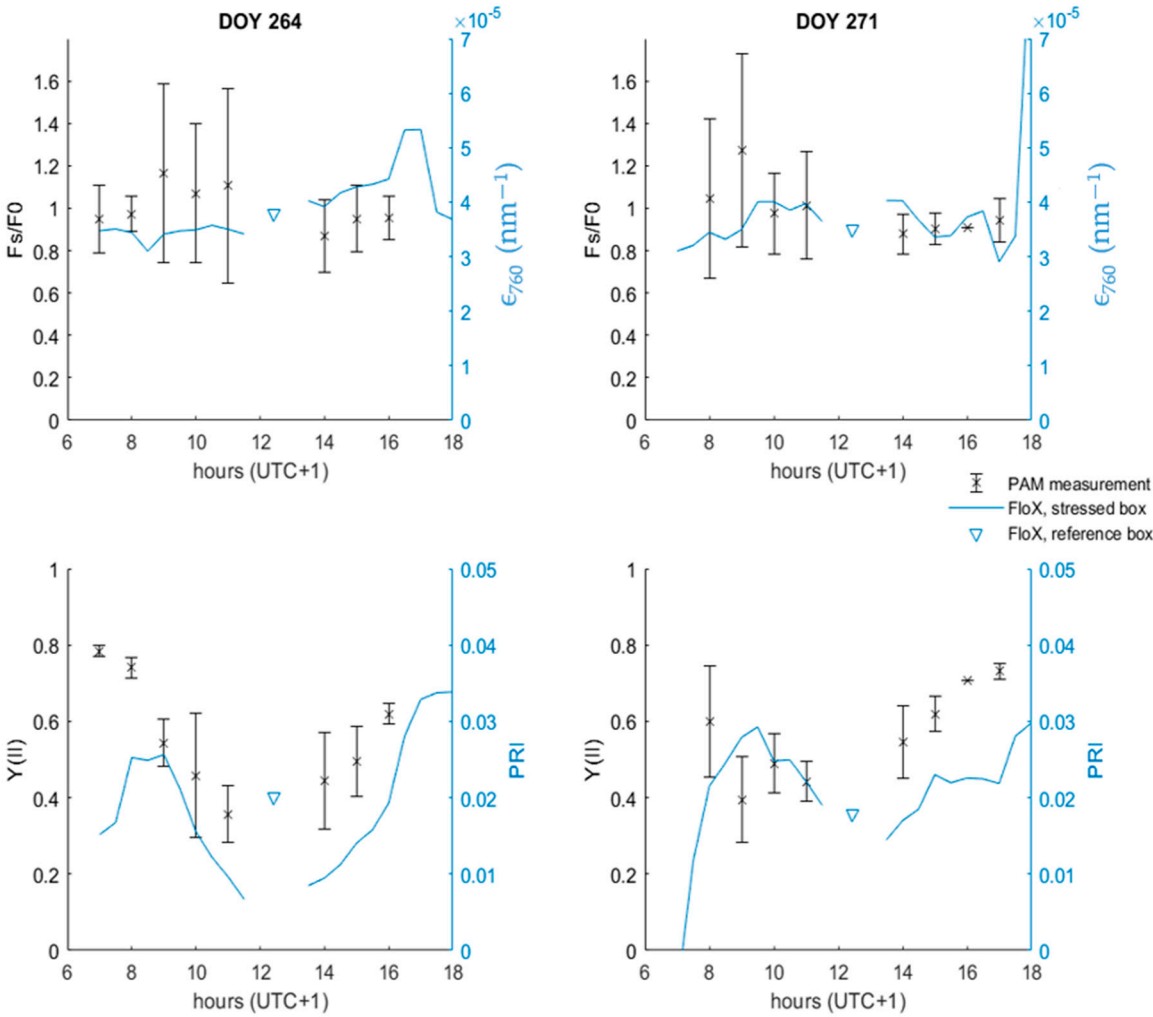

**Figure 9.** Behaviour of the PAM-based variables Fs/F0 and Y(II) as well as two FloX-based variables, $\epsilon_{760}$ and PRI for DOY 264 and DOY 271 during the lettuce experiment. Each PAM data point is an average of 10 measurements. The PAM measurements were taken in the stressed box.

## 4. Discussion

### 4.1. Relative Timing of the Plant Structural and Biochemical Reaction

In both the lettuce and mustard datasets, the $\epsilon_{760}$ reacted without delay to changes in light or water limitation (Figure 8). The decrease in $\epsilon_{760}$ was more expressed during the mustard experiment compared to the lettuce experiment. The lettuce, however, showed a simultaneous reaction in both its biochemical variable $\epsilon_{760}$ and in its structural variable, FCVI. A possible explanation for the different behaviour between the two crops lies in the difference in their degree of isohydricity. Because of the loose stomatal regulation by the lettuce plant [33], it is expected to lose its turgor more quickly compared to mustard. These variations in turgor induce a change in the leaf angle [34]. Field-studies have observed that these changes occur before any irreversible damage occurs, through necrosis, for example [35]. Near-infrared reflectance is therefore a promising early water stress indicator. A sub-daily temporal resolution is recommended here, since the changes in turgor pressure typically show a clear diurnal pattern [36]. Mustard's more isohydric stomatal regulation could explain its little diurnal FCVI variability. This suggests that the short-term changes in canopy structure are mainly useful as a stress indicator over anisohydric plants.

### 4.2. Plant Physiological Interpretation of the $\epsilon_{760}$ and PRI Behaviour

PAM-based studies have observed a decrease in fluorescence yield $\varphi_f$ in the case of high-light conditions with increasing light saturation [37,38], similar to the decrease in $\epsilon_{760}$

for high PAR values in Figure 4. Other PAM studies have observed a decrease in steady-state fluorescence (Fs/F0), such as in the case of water stress and high-light conditions, with a minimal around noon [13]. This behaviour is similar to the anticorrelations between $\epsilon_{760}$ and PAR shown in Figure 7. These findings confirm that $\epsilon_{760}$ can be conceptualized as a canopy-averaged variant of $\varphi_f$. The large standard deviations in Figure 9 point out that there is a large variability in leaf-scale $\varphi_f$.

Interpreting SIFB in physiological terms is harder compared to SIFA because SIFB structural and biochemical components cannot be separated. The light response curves in Figure 7 report a clear difference in light response between SIFA and SIFB; the SIFB seems to adhere to its linear relationship with PAR, even under high-light and low-water conditions, whereas SIFA emissions seem to saturate. Whereas various sources have observed some sort of decrease in SIFA during a drought stress [22–24,39], studies linking SIFB to drought stress have been inconclusive, with some reporting an increase [23] and others a decrease [22]. Leaf-scale studies report a decrease in SIFB emission in the case of a drought stress [40]. Figure 7 suggests that SIBY becomes anticorrelated with PAR, similar to $\epsilon_{760}$, albeit only in more severe stress conditions.

### 4.3. Use of SIF as a Water Stress Monitoring Tool

Because of its mechanistic link to the photosynthetic apparatus, $\epsilon_{760}$ is reactive to ambient stress conditions, being a combination of the soil water availability, the VPD and the PAR. These conditions change during the day. At dawn, when PAR approximates 0, a plant grows in a light-limited regime, regardless of its soil water status. Assuming cloud-free conditions, PAR and VPD follow a bell-shaped pattern, and the plant will switch to a water-limited regime once the PAR and VPD reach a certain level. The point at which this happens depends on the plant water status, and, therefore, also on the soil moisture as well as on the plant isohydricity. Figures 5–7 have shown that more intense water limitation results in a more expressed decrease in $\epsilon_{760}$, making it an indicator for the instantaneous stress conditions. By linking instantaneous stress to environmental factors, measurements of the degree to which a plant is in a water-limited regime ($\epsilon_{760}$) provide sufficient information to correct crop growth models for drought stress [24]. This makes $\epsilon_{760}$ an especially interesting indicator in the light of the upcoming FLuorescence EXplorer (FLEX), as it will provide global maps of $\epsilon_{760}$ with a spatial resolution of 300 m. Global-scale SIF from the TROPOMI sensor has already proven its value to classify ecosystems as water- or light-limited [41]. The drought-induced decrease in $\epsilon_{760}$ is expected to allow monitoring the intensity of the water limitation.

### 4.4. Use of PRI as a Stress Indicator

Figure 9 shows that the PRI neatly follows the same diurnal behaviour as Y(II), which is, by definition, inversely related to NPQ, making it a biochemical indicator that is complementary to SIF. When using PRI, it is important to consider that PRI bears both structural and biochemical information. Unlike to SIFA, there is no corrective vegetation index to separate the biochemical and structural components of PRI. This makes the interpretation of PRI in physiological terms less straightforward. Still, PRI time series revealed some stressed periods, as PRI did not adhere to its upward trend during this period. Using high-temporal resolution data of PRI, several methods using de-trending techniques exist to remove the structural effect from the PRI measurement [42,43]. The differences in the stress reaction between PRI and SIFA indicate that both variables are sensitive to different parts of the photosynthetic apparatus. This suggest that they provide complementary information on the plant's photosynthesis. The combination of ecosystem-scale PRI and SIFA has already proven its value for photosynthesis estimations [44].

### 4.5. Complementarity of Structural and Biochemical Information

Water losses through transpiration might result in a drop in turgor pressure. This induces changes in the leaf configuration, affecting the $f_{esc}$ and fPAR$_{chl}$. The plant's iso-

hydricity determines whether it will rather drop its turgor pressure or slow down its photosynthesis [45]. The drop in turgor pressure and the structural changes associated with it take place at the diurnal timescale [46], which is why diurnal observations of the FCVI (or any other structure indicator) are necessary in order to link canopy structure to drought stress. As, the $\epsilon_{760}$ is only sensitive to the biochemical component of SIF emission, FCVI and $\epsilon_{760}$ provide independent, yet complementary information on the instantaneous stress conditions. It is interesting to note that detecting the structural reaction requires less strict requirements regarding spectral resolution compared to SIF observations, but measuring the full diurnal cycle of FCVI is needed to link it to leaf angle and water stress changes. Because of the absence of full diurnal measurements, satellite-based SIF might be more reactive than satellite-based vegetation indices [47], despite the structural and biochemical reaction taking place simultaneously.

It is worth noting that water limitation also impedes vegetation growth, lowering its green biomass, which, in turn, affects various remote sensing metrics, including the NIR reflectance, and thus the FCVI. Short- and long-term variations in FCVI therefore represent other processes. The combined long- and short-term dynamics, affecting FCVI are visible on Figure 6; the upward trend in FCVI represents the growth of the lettuce plant, whereas the diurnal variations in FCVI are due to changes in the leaf angle.

## 5. Conclusions

This paper describes the reaction of the sun-induced chlorophyll fluorescence (SIF) emission and the canopy reflectance to increasing water limitation for both a mustard and a lettuce stand during a field experiment that lasted their entire growing seasons. The SIFA emission was decomposed into a biochemical component, being the fluorescence emission efficiency ($\epsilon_{760}$) and a structural component, being the Fluorescence Correction Vegetation Index (FCVI). Such operation was not done for SIFB. In addition, the photochemical reflectance index (PRI) was monitored. Both plants showed a biochemical reaction to increasing stress, most notably in the fluorescence emission efficiency ($\epsilon_{760}$). The reaction of $\epsilon_{760}$ to increasing stress was observed in two ways. First, $\epsilon_{760}$ decreases according to the stress intensity. Second, $\epsilon_{760}$ and PAR show an anticorrelation under water-limited days and a positive correlation under light-limited days. Under intensely stressed days, SIFBY and PAR also showed an anticorrelation. Given the more isohydric nature of the mustard plant, it showed a clearer biochemical reaction compared to the lettuce.

In addition to the biochemical reaction, the lettuce also showed a change in its canopy structure. The structural reaction is visible through diurnal variations in the FCVI. This behaviour was not observed for the mustard FCVI. Given the anisohydric nature of the lettuce plant, it was expected to show more variations in its turgor pressure compared to the mustard. This is consistent with the large diurnal variation in leaf angle by the lettuce canopy. In addition to the SIF, PRI (photochemical reflectance index) decreases in the case of stress. Affected by both the plant development and the leaf biochemistry, a high temporal resolution of PRI measurements is needed to unravel the leaf biochemistry and plant development effects. The findings in this paper contribute to the monitoring of plant water status. Whereas this paper explains the reaction of the leaf biochemistry and the canopy structure at the local scale, the same logic can be applied at larger scale.

The structural reaction (FCVI) is a valuable stress indicator if the seasonal and daily dynamics in the canopy can be disentangled. The seasonal dynamics are influenced by vegetation growth, whereas the short-term dynamics are related to changes in leaf angle, which are influenced by the turgor pressure. The biochemical reaction is a decrease in $\epsilon_{760}$ during periods of a low soil water availability and high water demand and is especially interesting information for the FLuorescence EXplorer (FLEX) mission. The photochemical reflectance index (PRI) provides additional information on the plant's biochemistry, but its dependency on the canopy structure hampers a clean interpretation of PRI in terms of leaf biochemistry.

**Author Contributions:** Conceptualization, S.D.C., H.V., P.D. and F.J.; Data curation, S.D.C.; Formal analysis, S.D.C.; Funding acquisition, S.D.C., H.V. and F.J.; Investigation, F.J.; Methodology, F.J.; Project administration, H.V. and F.J.; Resources, H.V. and F.J.; Software, F.J.; Supervision, P.D. and F.J.; Writing—original draft, S.D.C.; Writing—review & editing, H.V. and F.J. All authors have read and agreed to the published version of the manuscript.

**Funding:** This research has been funded by the Fonds pour la Formation à la Recherche dans l'Industrie et dans l'Agriculture (FRIA, Belgium) and the Deutsche Forschungsgemeinschaft (DFG, German Research Foundation) under Germany's Excellence Strategy—EXC 2070—390732324.

**Data Availability Statement:** The data are available upon request.

**Acknowledgments:** We express our gratitude to Robin Van Tendeloo for the box sketches and to Thomas Meyer, Jordan Bates and Felix Bauer for the help during the fieldwork.

**Conflicts of Interest:** The authors declare no conflict of interest.

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
