# Peer review of "Remote Sensing of Instantaneous Drought Stress at Canopy Level Using Sun-Induced Chlorophyll Fluorescence and Canopy Reflectance"

_remotesensing, doi:10.3390/rs14112642_

Round 1

Reviewer 1 Report

In this study, the authors used the in-situ measurement (SIF and reflectance) data to study the instantaneous drought stress at the canopy level. Two kinds of vegetation showing different responses to drought are selected in this study. For the results, the authors analyzed and discussed them from plant structural and biochemical. Some indicators show a promising ability to study vegetation stress. The experiment is well designed and the results are very interesting. I would like to recommend accepting with minor revision.

Minor comments:

The authors should check the equation number and the format.

Line 58-60; Line 130; Line 165.

Author Response

Dear Sir/Madam

Thank you for having taken reviewed my paper. We provided a bit more context in the introduction, re-wrote some parts of the discussion and adapted the numbering of the equations. 

Best wishes

Simon

Reviewer 2 Report

This paper designed experiments to detect SIF and surface reflectance of two plants with different isohydric degrees. Then the manuscript analyzed the data and evaluated different parameters in indicating drought stress. But major concerns about the manuscript are:

  1. The authors should state previous case studies on SIF monitoring drought stress in Introduction section.
  2. To measure SIF on drought stress, it is better to take or create at least one indicator to measure the relationship between drought (VPD or pF) and vegetation parameters (in this manuscript such as SIF, PRI, etc). Since the author measured many parameters for a long time, it is difficult to distinguish a better drought-indicators from other indicators based on dense scatters or lines in present manuscript.

Besides, some minor concerns are listed:

  1. Line 47, “CO2” should be “CO2”.
  2. Line 59, there exists an unexpected Enter space.
  3. Line 113, there should be at least one reference of “USDA textural classification”.
  4. Line 167, the manuscript misses the sequence number of the second equation.
  5. Line 200 and 201, there exist spelling errors of “°C”.
  6. Line 227, in Figure 4, there should be other parameters such as soil moisture or pF in the plots to indicate high and low water stress.
  7. Line 232, the sequence number of the sub-sections should be organized.
  8. Line 324, the authors should state which figure in the manuscript showing the results.
  9. Line 367, as for section 4.3 and 4.4, it is actually hard to compare those parameters (ε760, SIF, PAR, etc) through several scatters, maybe the authors should provide some evaluation indicators to measure various parameters’ results.

Author Response

Dear reviewer

Thank you for your comments. Please find my answers below

Best wishes

Reviewer 3 Report

This research article analyzed the impact of drought stress on SIF emission and surface reflection for mustard and lettuce plants through a breakdown of the SIFA emissions to structural and biochemical components. The authors do an excellent job providing extensive background context, a scientifically sound description of methods, and and a thorough presentation of results and discussion. I recommend acceptance of the present article after a few very minor revisions for grammatical mistakes or small clarifications. In my time of reviewing articles, this has been one of the best articles I have reviewed. Thank you for the pleasure off reading it!

Line 59- strange break

Line 126- the parentheses should be moved after the acronym

Lines 153-154 What is the spectral resolution of the hyper spectral images? you focus on the temporal resolution a good deal but don't share that information.

Section 2.4- I believe equations 2 and 3 are referred to as equations 1 and 2 before then jumping to equation 4.

Line 197- a random period after the title

Lines 252-255- awkward phrasing

Author Response

Dear sir/madam

Thank your for your appreciation and your comments. Please find our answers here in attachement.

Best wishes

Round 2

Reviewer 2 Report

all problems have been revised.